# Optimization and Experiment of Livestock and Poultry Manure Composting Equipment with Vented Heating

Kailin Ren [†], Lide Su [†], Yong Zhang *, Xiang He and Xuyang Cai

College of Mechanical and Electrical Engineering, Inner Mongolia Agricultural University, Hohhot 010018, China; m17362908182@163.com (K.R.); sld@imau.edu.cn (L.S.); h17513207697@163.com (X.H.); cxy153@proton.me (X.C.)

* Correspondence: yongz@imau.edu.cn; Tel.: +86-136-7474-8009

[†] These authors contributed equally to this work.

**Abstract:** As livestock and poultry farming expands in China, the generation of fecal waste has significantly increased. Inadequate waste management can contribute to environmental pollution. This study seeks to optimize small-scale composting systems to address the inefficiencies and the limited automation of traditional composting equipment. We redesigned the mixing blades and refined the ventilation heating system, establishing an efficient mixing mechanism and an energy-saving ventilation heating system. A control system, incorporating Monitor and Control Generated System and Programmable logic Controller, was developed for real-time monitoring and adjustment capabilities, substantially enhancing automation levels. The optimization proved effective by reducing the composting cycle from 13 days to 11.5 days, increasing the GI value from 83% to 89%, and lowering the humidity from 8.9% to 8.1%.

**Keywords:** composting equipment; mixing blade; ventilation heating device; control system; sheep manure

## 1. Introduction

Since ancient times, China has been a major country for animal husbandry. The continuous population growth and increasing demand for agricultural products have driven the development of planting and animal husbandry toward intensification, scale, and industrialization [1–3]. Consequently, the number of livestock farms and the amount of feces produced have also increased. Animal manure and crop straw, accounting for over 95% of organic solid waste, are the two main sources in China [4]. By 2020, livestock and poultry manure pollution reached 298 million tons [5]. Among the major livestock and poultry breeding provinces, Inner Mongolia has the country's highest cattle and sheep stock, maintaining stable annual growth and indicating potential in resource utilization [6,7]. Effectively treating animal manure could enable traditional small-scale farms, currently restrained by their inability to handle large quantities of it, to expand their operations. Such expansion of aquaculture farms, facilitated by efficient manure management, would significantly benefit China's aquaculture industry.

Animal manure is both a rich source of organic matter and a potential environmental pollutant, containing pathogenic bacteria and nutrients which can contribute to water eutrophication and soil contamination if improperly treated [8,9]. Recognizing these challenges, China has outlined goals for agricultural sustainability to ensure the comprehensive utilization of livestock waste and crop straw by 2030 [9–11]. Given the urgency of harmless manure treatment, composting—a widely used method—transforms livestock and poultry manure into organic fertilizer, increasing agricultural yield and mitigating soil degradation from traditional fertilizers [12–16]. However, uncontrolled factors such as temperature, humidity, and ventilation during natural fermentation composting can lead to incomplete decomposition, resulting in wasted resources, environmental pollution, and ineffective composting [17–19]. Therefore, the use of specialized composting equipment emerges as a

more effective solution [20]. Differences arise when comparing the natural or static composting process to composting utilizing equipment. For instance, employing composting equipment facilitates external heat sources to expedite the heating of the pile material. This technique significantly shortens the composting time by reducing the heating period compared to natural or static composting. Therefore, the equipment-aided composting process is condensed into three stages: the high-temperature stage, the cooling stage, and the maturity stage.

Extensive research has been conducted globally on the structural components and control mechanisms of composting treatment equipment, as well as post-processing techniques. Shyan et al. [21] and Makan et al. [22] developed structurally optimized composting reactors, but these innovations lacked sophisticated control systems, presenting limitations in real-time monitoring and automation levels. Xiong et al. [23] optimized compost temperature and maturity through an intelligent reactor, but the control system was not integrated, resulting in insufficient automation. Efforts to integrate control functionality into composting equipment, such as those performed by Jonas Schneider et al. [24] and Yang Y. et al. [25], have advanced compost treatment capabilities, but significant challenges remain due to deficiencies in control system design. Jain et al. [26] designed a rotary drum composting machine, achieving ideal mixing effects in aerobic composting of waste such as water hyacinth, vegetable waste, sewage sludge, and industrial sludge. Yu et al. [27] and Li et al. [28] designed composting systems with automatic control systems to manage temperature, mixing, and ventilation. However, these lacked efficient mixing structures, compromising the quality of composting results. The new high-temperature aerobic composter developed by Yan et al. [29] includes four parts: control panel, fermentation tank, air compressor ventilation system, and ammonia absorption system. Concurrently, researchers have adopted advanced techniques during the post-treatment phase of experiments. Researchers, including Davide Papaurello [30] and B. Scaglia et al. [31], utilized innovative methods for post-processing, such as using direct injection mass spectrometry (PTRMS). These studies serve as a basis for developing comprehensive composting equipment with balanced structural designs and control systems, as well as efficient post-treatment techniques. These improvements aim to enhance composting efficiency, further contributing to waste management and sustainable agriculture.

In conclusion, while scientific advancements have been realized in the research on structure, control, and result detection of organic waste composting equipment, these studies have primarily focused on structure or control, with little emphasis on assessing the relationship between the composting equipment structure and control systems. To holistically address the challenge of harmless treatment of livestock and poultry manure and crop straw, it becomes essential to design composting equipment offering efficient mixing and high levels of automation. This article contextualizes this topic by optimizing and improving the mixing effects and control system based on existing small-scale aeration heating animal manure composting equipment [32]. The primary objective is to develop equipment with excellent composting effects, high automation levels, and real-time monitoring capabilities. Such equipment ensures compost maturity and fermentation rate, aspects that are critical for the development of China's compost industry.

## 2. Experiments and Methods

### 2.1. Overall Structure and Workflow of the Equipment

2.1.1. 3D Model of Composting Equipment

Figure 1 shows the overall structure of a small-scale composting aerated heating livestock manure composting equipment. The composting equipment is modeled with four main parts: the composting fermentation system, aerated heating system, control system, and rack. The composting fermentation system, also called the fermentation silo, consists primarily of fermentation tanks, mixers, reclaiming devices, and deodorization devices. The ventilation heating system, also known as the hot gas device, includes an air pump, heating box, electric heating pipe, and ventilation pipeline. The control system is made

up of a control box and a sensor section, featuring a touch screen, programmable logic controller, and other components. Lastly, the rack is comprised of a base plate, aluminum profiles, and fasteners.

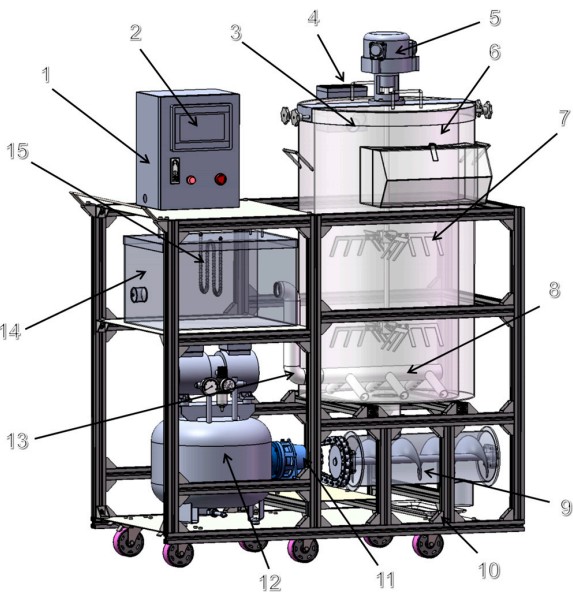

**Figure 1.** Three-dimensional model of composting equipment. 1. Control box, 2. Touch screen, 3. Exhaust pipe, 4. Deodorization device, 5. Pneumatic agitator, 6. Fermentation tank, 7. Mixing blade, 8. Aeration pipe, 9. Screw conveyor,10. Frame, 11. Cycloid reducer, 12. Air pump, 13. S-shaped ventilation pipe, 14. Heating box, 15. Electric heating tube.

### 2.1.2. Working Process

Figure 2 illustrates the workings of the composting equipment as animal manure is processed inside the fermentation tank and the stirring paddle operates. Initially, as the stack enters the tank at room temperature, the heating tube activates, followed by the air pump supplying ventilation. Throughout the composting process, microorganisms break down organic matter, consuming oxygen and reducing the oxygen concentration in the storage tank. The control system maintains the oxygen concentration within an optimal range. To expedite the decomposition of organic material and enhance the fermentation rate, it is essential to sustain a composting temperature between 55 and 65 °C [32–35]. If the stacking temperature diverges from this range, the control system regulates the electric heating pipe and the ventilation apparatus, ensuring a consistently appropriate stacking temperature.

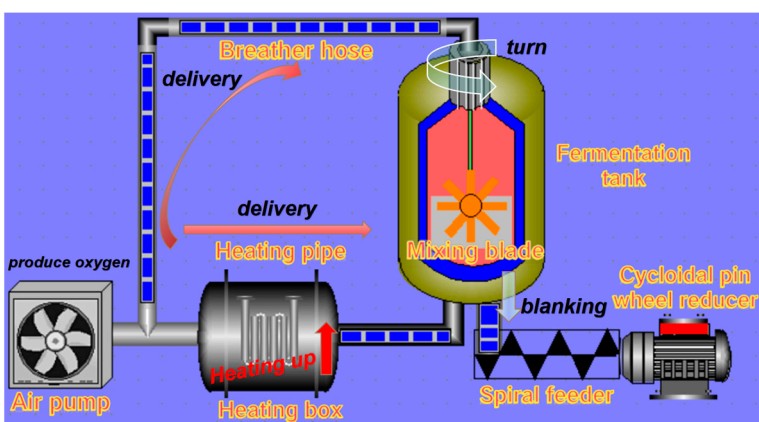

**Figure 2.** Working principle diagram of composting equipment.

### 2.2. Structural Optimization of Ventilation Heating System

The original composting device considered setting up an insulation layer for the fermentation tank but did not provide insulation treatment for the heating box. In this section, by calculating the heat loss inside the box, the heating tube was re-selected, and the heating box was insulated. Complete a ventilation heating system that is more energy-efficient and has better heating effects than the original ventilation heating system.

The air generated by the air pump enters the heating box through a pipeline and then flows through an electric heating pipe to be heated. During this process, the air absorbs heat and heats up while also experiencing various forms of heat loss. Therefore, it is necessary to conduct a thermal balance analysis of the process to understand the heat transfer during air heating in order to determine the appropriate power of the electric heating tube and then select the model and size based on the commonly used electric heating tube types on the market. During the air heating process, the heat balance equation is as follows:

$$q_{eht} = q_{ia} + q_{hc} + q_{ste} + q_{else} \tag{1}$$

The heat released by the electric heating tube $q_{eht}$.

Assuming the power of the electric heating tube is P (W), the heat released by the electric heating tube can be expressed as

$$q_{eht} = 3.6P \text{ kJ/h} \tag{2}$$

The heat absorbed by the air pump during heating up $q_{ia}$:

$$q_{ia} = c_{air} \cdot \phi_{air} \cdot \Delta T_{air} \cdot \rho_{air} \tag{3}$$

Based on the known calculation results [32], it can be concluded that the required ventilation rate for the stacking material is at least 30 L/min, so the ventilation rate output by the air pump on the ventilation line of the heating box should meet this requirement, so ϕ air is taken as 30 L/min, and after converting to units, it is 1.8 m/h. The density of air and constant pressure specific heat capacity at 15 degrees Celsius under 1 standard atmospheric pressure are 1.226 kg/m$^3$, 1.005 kJ/kg·K. The change value of air temperature is 65 K. By substituting each parameter into Formula (3), it can be obtained that the heat absorbed by the air pump during heating is

$$q_{ia} \approx 144.159 \text{ kJ/h}$$

Heat conduction on the wall of the heating box:

$$q_{hc} = \lambda_{air} \times A \times \frac{\Delta T}{d} \tag{4}$$

Due to the stainless steel wall's good thermal conductivity, the temperature of the heating box wall, without insulation treatment, is approximately equal to the internal hot air temperature. A large temperature difference between the inside and outside results after heating, increasing heat loss and reducing the heating efficiency of the heating tube. Research has shown that the composting reactor's side walls contribute far more to heat loss than other parts. During the composting process, the side walls and bottom surfaces of the fermentation tank that come into direct contact with the pile material cause significant heat loss due to heat transfer [36].

The heating box is insulated with a 5 mm aluminum foil polyurethane insulation board to address this issue, as shown in Figure 3. The polyurethane material has a stable structure and excellent sound absorption and insulation properties. Under normal maintenance and use, the lifespan of hard foam polyurethane insulation can reach several decades or more [37].

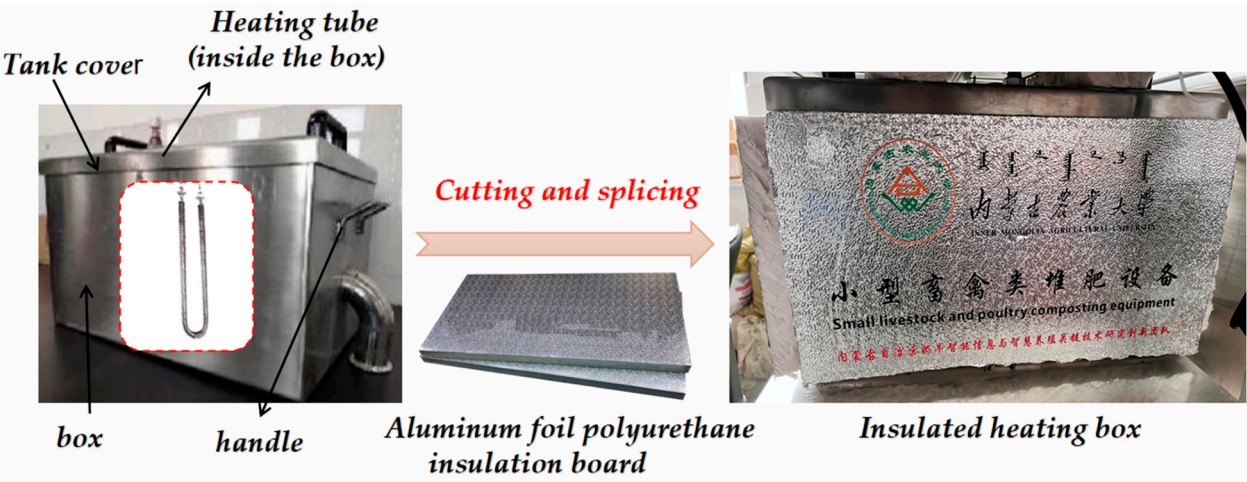

**Figure 3.** Heating box insulation treatment.

The maximum temperature difference between the heating box wall and the external air in Inner Mongolia is 50 K. The average value of 25 K is taken for calculation as the temperature difference between the two gradually increases from 0 to 50 K. The thermal conductivity $\lambda_{air}$ of normal temperature air at 1 standard atmospheric pressure is 0.025 W/m K; The contact area between the heating box and the external air is 0.685 m²; Due to the wall thickness of the heating box being 3 mm, the distance d for heat transfer on the wall is 1.5 mm, which is 0.0015 m. By substituting the above parameters into Formula (4), the heat conduction on the wall of the heating box can be obtained as

$$q_{hc} \approx 285.417 \, W = 1027.501 \text{ kJ/h}$$

The heat absorbed by the evaporation of water in the air $q_{ste}$

$$q_{ste} = f_{air} \times \rho_{air} \times \Delta x \times q_l \tag{5}$$

By consulting relevant materials and using computer-aided tools such as air physical parameter calculators, it can be concluded that in Dangkong, when the air temperature is 288.15 K and the humidity is 20% (taking Inner Mongolia as an example), the net mass $\Delta x$ of water vapor in the air is about 0.002 kg/kg. Similarly, under this condition, the enthalpy value of water vapor is approximately 20.4 kJ/kg. By substituting the corresponding parameters into Formula (5), the heat absorbed by water evaporation in the air can be obtained as

$$q_{ste} \approx 0.095 \text{ kJ/h}$$

Other forms of heat loss $q_{else}$.

In the actual air heating process, in addition to the aforementioned heat loss methods, there are also some more complex forms of heat transfer, mainly including thermal radiation and heat dissipation at the connection of ventilation pipes. Therefore, in this section of the heat balance analysis, it is collectively referred to as other forms of heat loss, represented by $q_{else}$.

Based on the calculation results of each heat loss form mentioned above, combined with the heat balance Equation (1), it can be concluded that

$$q_{eht} \geq q_{ia} + q_{hc} + q_{ste} \tag{6}$$

$$3.6P \geq 144.159 + 1027.501 + 0.0095,$$

Further, the power of the electric heating tube can be obtained as

$$P \geq 317 \text{ W},$$

Based on the above calculation results and considering the heat loss from other sources during the heating process, along with the power specifications and types of electric heating tubes commonly available in the current market, we select the size parameters and models of the electric heating tubes. These criteria are guided by the structural dimensions of the heating box. Ultimately, the power of the electric heating tube used in the hot gas device is determined to be 450 W, while the chosen type is a W-shaped finned dry-burning electric heating tube (working voltage 220 V). The electric heating tube's positive and negative electrodes have a center distance of about 300 mm, and the tube body diameter (including fins) is about 24 mm.

After installing the insulated heating box and selecting heating pipes, we recorded the time required for the temperature to rise and reach 80 degrees Celsius using a high-precision digital thermometer. We then compared the results to verify the reliability of the insulated ventilation heating system. It is crucial for the hot air temperature flowing into the fermentation tank to be above 80 °C, ensuring that the effective temperature of the compost reaches the specified range. The time taken for the temperature to reach 80 °C is considered the actual preheating time.

Figure 4 demonstrates the effect of implementing insulation measures on the heating system. The time needed for the temperature inside the heating box to reach above 80 degrees Celsius has decreased from 31 s to 29 s, a reduction of 2 s. Notably, during the initial 17 s, the insulation board's capacity to prevent heat dissipation was relatively modest. However, after this point, the temperature difference between the interior and exterior became more substantial due to the insulation measures in place. The power consumption of a heating tube is evaluated based on the energy expended to reach the target temperature before and after the improvement. As per calculations, the energy usage of the improved heating tube has declined from the original 1.55 KJ to 1.31 KJ. Consequently, the improved heating system accomplishes the task of reaching 80 degrees Celsius more efficiently. These comparative results highlight the enhanced heating section's efficacy and energy-saving qualities.

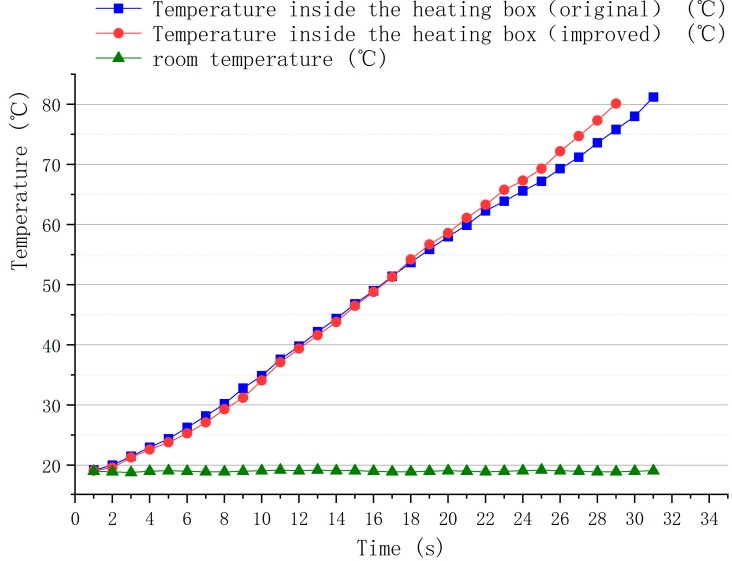

**Figure 4.** Comparison of preheating time test results for heating boxes.

### 2.3. Optimization of Mixing Blades

As indicated in prior research [38,39], the maximal deformation of the 3-blade E-type stirring impeller occurs at the tip of the E-type blade. Meanwhile, the peak stress and elastic strain reside at the connection between the blade and the fixed component, reaching a maximum stress of 58.194 MPa. The prolonged continuous operation could potentially cause the connection between the blades and fixed components to break, which

would subsequently impact the normal functionality and delay the progress of aerobic composting fermentation.

To address this issue, we designed a new structure for the mixing blade that focuses on the potential fracture points at the connection between the fixed end and the blade. This new design is shown in Figure 5.

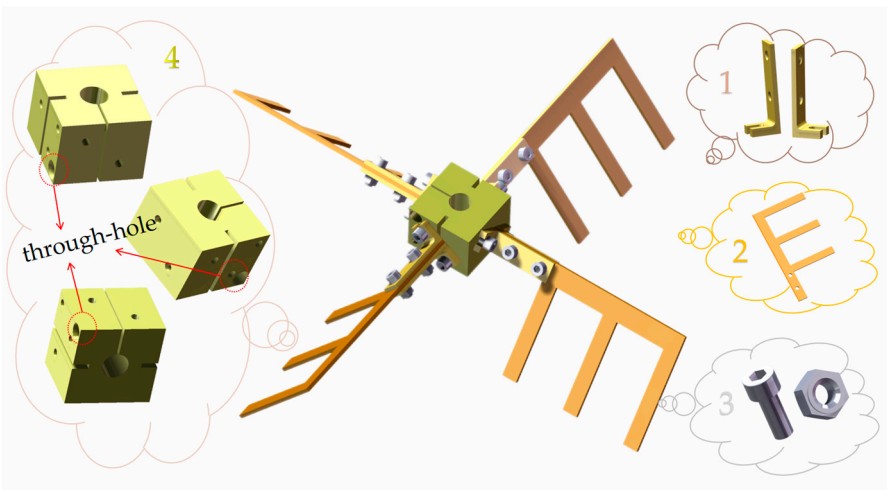

**Figure 5.** Improve the structural diagram of the 4-blade stirring E-shaped blade. 1. L-shaped fixing parts; 2. Mixing blade; 3. Screws and nuts; 4. cubic fixing part.

The three-dimensional model of the reactor was established using SOLIDWORKS 2023 software. Boolean operations were employed to divide the action and static regions of the stirring device, with the stirring blades and a portion of the stirring shaft designated as the moving region and the remaining portion as the static region. The 3D model of the mixing device was then imported into Mesh for grid division, and the sliding grid method was applied to set the rotating and fixed areas, simulating the mixing blades' motion within the tank. The model adopts an unstructured tetrahedral mesh, which is divided into two parts for calculation: fixed region and rotating region. The results obtained through simulation are illustrated in Figure 6.

As demonstrated in the simulation results in Figure 6, the maximum deformation point, maximum stress point, and maximum elastic strain point all occur at similar positions. However, their overall values have decreased significantly. For instance, the maximum pressure drops to 41.994 Mpa, with the maximum pressure point being the L-shaped fixed plate. If the L-shaped fixed plate is damaged due to prolonged work, it can be easily replaced to ensure the composting equipment continues to operate normally. In contrast, if the original mixing blade is damaged or broken, the adhesive needs to be removed, and the entire mixing blade replaced, which is an extremely complex dismantling project. The results show that the improved mixing blade structure enhances the reliability and convenience of the entire device, significantly improving economic efficiency. Manufacture and assemble the designed mixing blades, as illustrated in Figure 7.

EDEM simulation is employed to analyze the stirring situation of materials during the composting process. By comparing the improved stirring blades with traditional blades, the difference in stirring effects between the two is analyzed, and the feasibility of the stirring blades designed in this article is evaluated. During the simulation analysis, it is necessary to simplify the equipment model and retain only key components such as the main body of the fermentation tank, rotating shaft, and mixing blades. The simplified model can meet the needs of simulation analysis while reflecting the actual stirring process of the pile material more realistically.

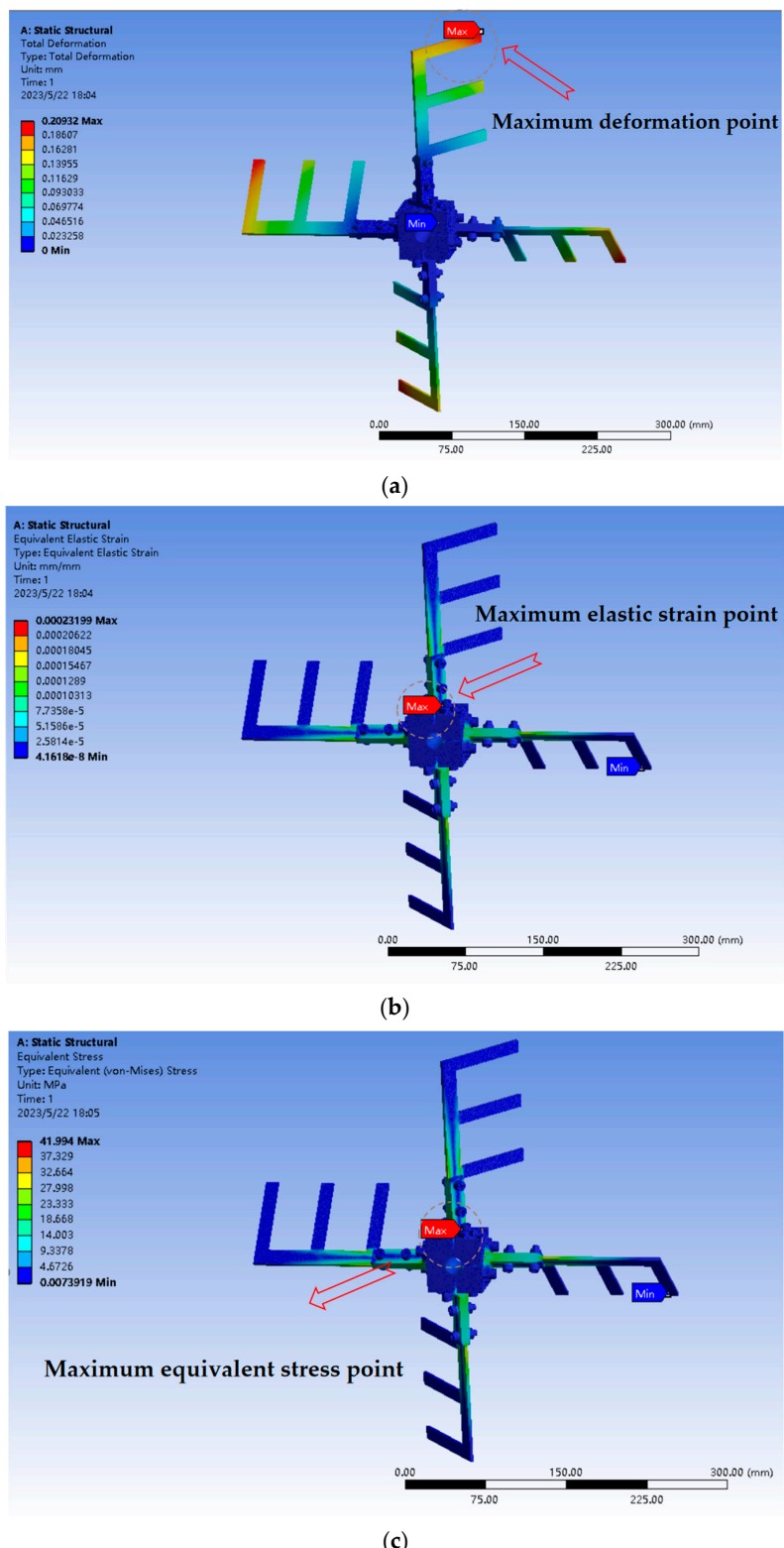

**Figure 6.** Four-blade E-shaped stirring blade diagram based on ANSYS. (**a**) Deformation analysis cloud chart of four E-shaped stirring blades; (**b**) Elastic strain analysis cloud chart of four E-shaped stirring blades; (**c**) Stress analysis cloud chart of four E-shaped stirring blades.

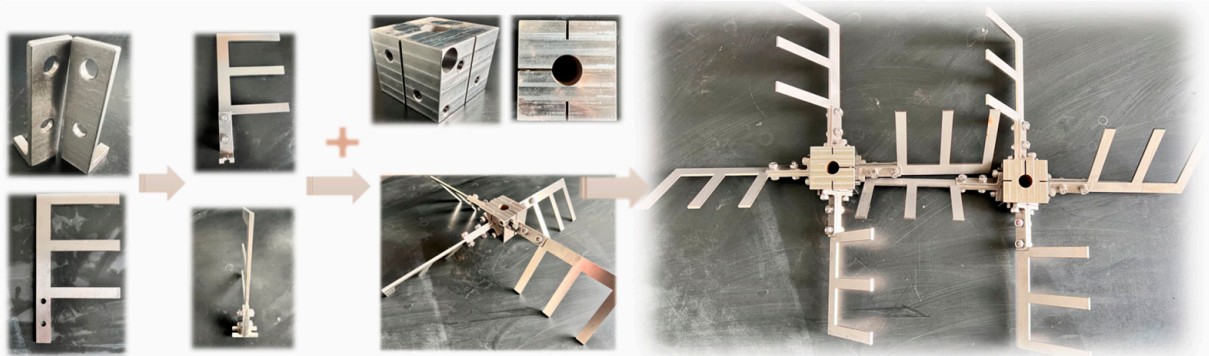

**Figure 7.** Improved 4-leaf E-blade entity diagram.

Taking the improved blade stacking and stirring model as an example, two types of sheep manure particle models were set: circular and elliptical, both accounting for 50% each. Due to the much smaller amount of corn straw added during actual composting compared to sheep manure, the influence of straw particles is ignored in the simulation, and only the flow performance of sheep manure particles is considered. Considering the need to add water to the material to adjust the initial moisture content during actual composting, the "Hertz Mindlin with JKR" model was chosen to simulate the contact properties between wet particles and between wet particles and geometric bodies. Taking particle motion at the 6.5 s mark during the simulation process as an example, a comparative analysis was conducted for three types of stirring blades. The simulation results are illustrated in Figure 8.

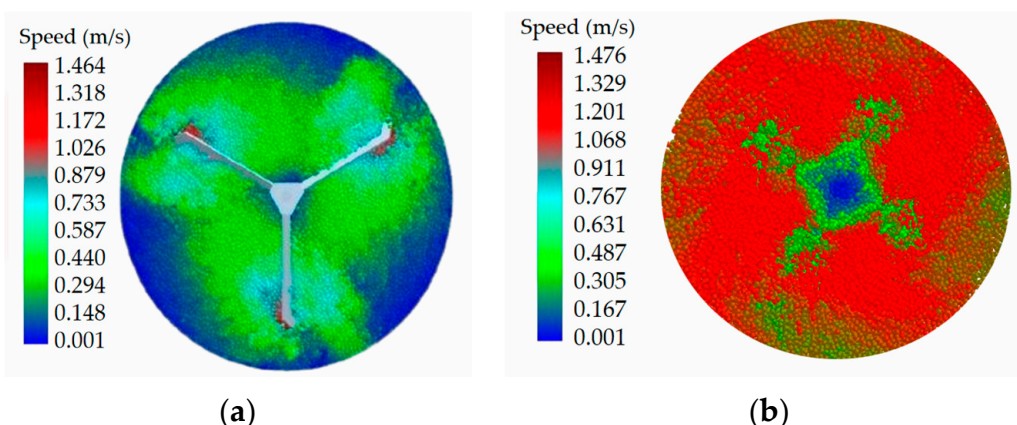

(a)                                          (b)

**Figure 8.** Motion speed image of particles at the 6.5 s in simulation. (**a**) Traditional blade; (**b**) Improved blades.

The improved blade body is designed to be transparent to facilitate the observation of particle velocity. Figure 8 clearly illustrates that under the influence of the improved blade stirring, both the particle motion speed and overall stirring range are significantly larger than those seen with unimproved versions, resulting in optimal stirring effects for the stacking material. In comparison to the unmodified three-blade design, certain areas at the edges are not effectively stirred during the mixing process. This issue is resolved in the improved blade design by increasing its radius, dividing it into four blades with 90-degree intervals, and adding 45-degree secondary blades. As particles at the leading edge of the blade move downwards, they are lifted along the leading edge due to the blade's action. With the help of the auxiliary blade, the particle velocity at the trailing edge of the blade increases, and peripheral particles are mobilized due to the interparticle collision. The synergistic effect of the upper and lower blades forms a mutual flow among particles, thereby enhancing the stirring performance of the stirrer. As a result, as shown in the figure,

the particle motion speed of the blades is relatively high, appearing in red. This design has spurred a large-scale composting movement, resulting in thorough overall mixing and maturation of the compost while reducing issues arising from uneven composting.

To collect unit time particle mass flow data in the area between two sets of mixing blades and indirectly reflect the number of particles and their fluidity in the middle of the pile under different types of blade agitation, set up a particle mass flow sensor. The setting area of the said sensor is represented by the grid section in Figure 9a.

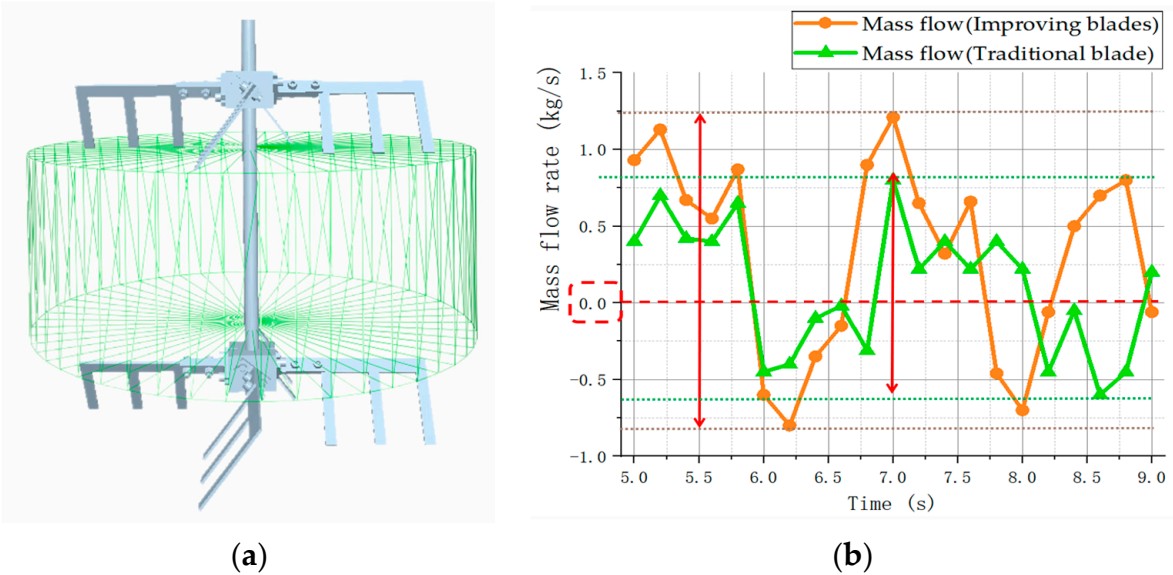

**(a)**  **(b)**

**Figure 9.** Particle mass flow statistics. (**a**) Schematic chart of particle mass flow sensor setup; (**b**) Broken line graph of particle mass flow in the corresponding areas of the two types of blades.

The mass flow of masses in the corresponding regions of the two blades was analyzed, and the results are shown in Figure 9b. The positive and negative signs in the figure indicate the opposite direction of particle flow, while the values represent the particle mass flowing through the set area in unit time. Under the improved blade stirring, particles in the middle of the pile exhibit enhanced fluidity, indicating a more frequent axial flow and a larger particle flow volume during the stirring process. In summary, the stirring blades designed in this study outperform traditional forms of stirring blades in terms of radial stirring performance and axial flow performance of the stack. This confirms the feasibility and rationality of the design.

### 2.4. Optimization of Control Part

The control objects of the composting control system include air pumps, electric heating pipes, and motors, while the control reference factors encompass the temperature of the stack material and the oxygen concentration in the tank. The system's goal is to manage the starting and stopping of the air pump and electric heating tube based on changes in stack temperature and oxygen concentration, as well as controlling the motor during the material retrieval stage. The composting control system features two control modes: automatic control and manual control.

In order to achieve the above functions, the control section will be optimized for PLC control and MCGS touchscreen programming control. Optimization of the control section will include both PLC control and MCGS touchscreen programming control. The composting control system primarily consists of a detection part and a control part. The detection part features soil temperature and humidity sensors, air temperature and humidity sensors, and oxygen concentration sensors. The control part is mainly composed of a touch screen, PLC, intermediate relay, AC contactor, switch power supply, air switch, and other components.

The wiring diagram of each component of the compost control system designed in this study is shown in Figure 10.

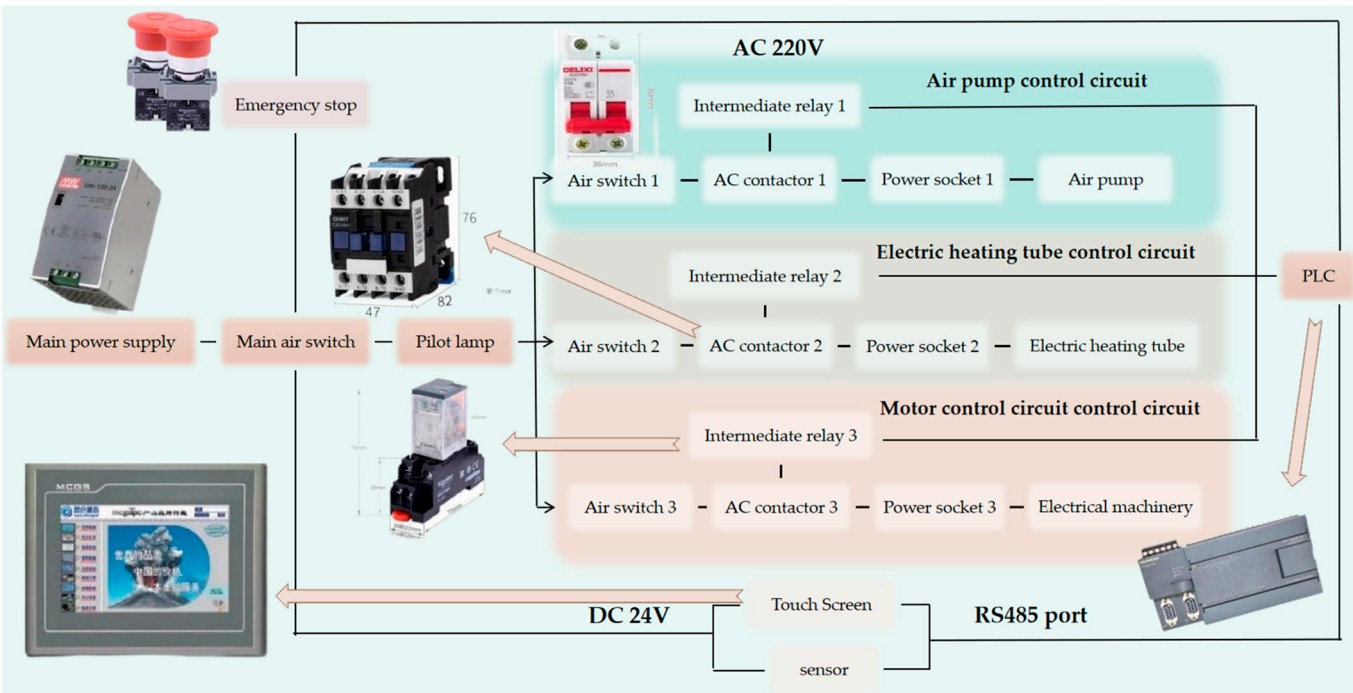

**Figure 10.** Wiring diagram and component diagram of compost control system.

The composting control system employs the Kunlun TPC7062Ti connected touch screen as the human–machine interface for the composting equipment. Detailed explanations for the login screen, monitoring screen, parameter settings, trend screen, report query, alarm query, and real-time database content in the user window will be provided using the MCGSE configuration environment software to edit and implement these functions.

The actual operation of the compost control system and the improved composting equipment is depicted in Figure 11, and all target functions can be achieved, and all electrical appliances can work normally.

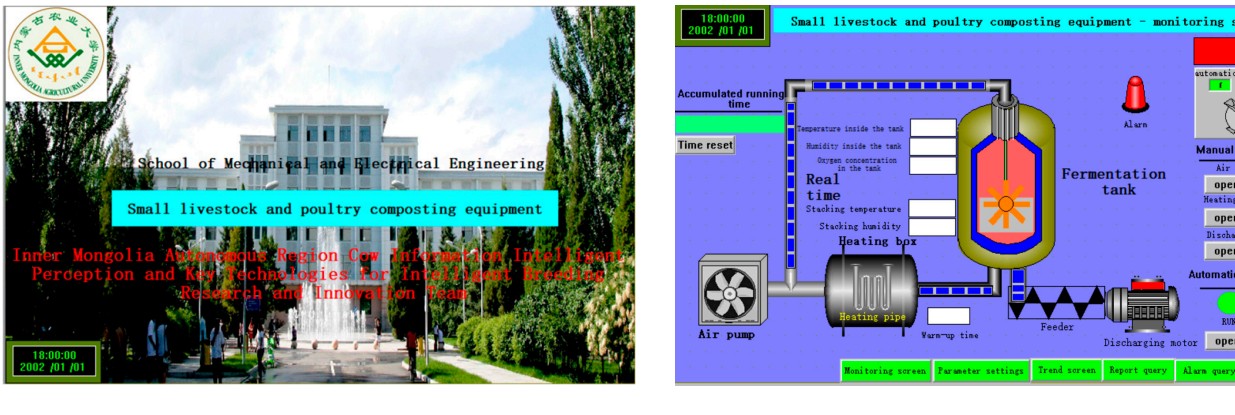

(**a**)         (**b**)

**Figure 11.** *Cont.*

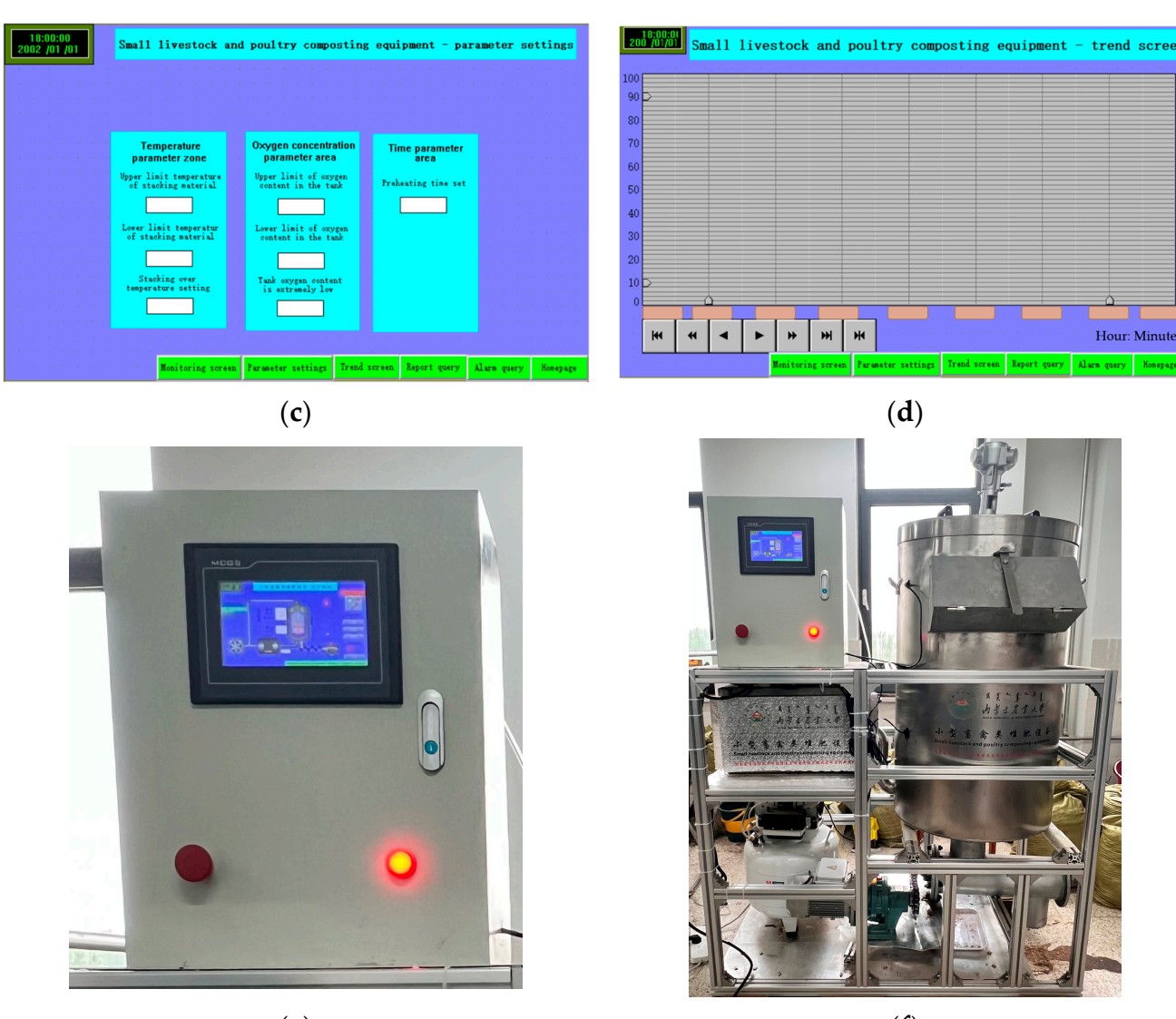

**Figure 11.** Normal operation of the control section. (**a**) Login screen; (**b**) Monitoring screen; (**c**) Parameter settings; (**d**) Trend screen; (**e**) External control box; (**f**) Improved composting equipment entity.

### 2.5. Experimental Verification of the Practicality of Composting Equipment

The Inner Mongolia Autonomous Region is abundant in sheep manure resources, leading to a particularly severe problem of manure accumulation. Simultaneously, Inner Mongolia is also a major distribution area for corn cultivation in China. After the corn harvest, a large amount of straw is produced, which contains rich organic matter that can enhance the compost's organic matter content and increase the final product's value [40–43]. Thus, sheep manure and corn straw were chosen as composting materials for the experiments.

The sheep manure used in the composting experiment was sourced from a fertilizer processing plant in Salaqi Town, Tumote Right Banner, Baotou City, Inner Mongolia Autonomous Region. The corn straw was obtained from the agricultural machinery laboratory of our college and comprised of crushed straw. A certain mass ratio is used to mix sheep manure and corn straw, and an appropriate amount of water is added to adjust the mixed pile's moisture content to 60%. Before composting begins, a sufficient quantity of fermentation agents should be added to the pile to increase the microbial communities' content and enhance the organic matter decomposition effect, thereby promoting pile maturity. The instruments utilized in the experiment are detailed in Table 1.

**Table 1.** Composting test instrument.

| Serial Number | Name | Function |
| --- | --- | --- |
| 1 | Electronic scale | Weigh sheep manure |
| 2 | Activated carbon | Adsorption waste |
| 3 | Seedling tray | Planting mung beans |
| 4 | Farmland soil | Mixed fermented compost for mung bean cultivation |
| 5 | Mung bean | Measuring germination rate |
| 6 | Starter and glass measuring cylinder | Promote compost maturity |
| 7 | Soil detector | Detecting factors such as moisture content, pH value, and conductivity of compost |
| 8 | barrel | Mix compost amount and moisture content |

To prepare the compost, mix sheep manure and straw in a 20:1 ratio [44–47]. Then, add water to adjust the moisture content to 60%, followed by the addition of Manure Fermentation starter. Manure Fermentation starter contains Bacillus subtilis, Streptomyces downer, Streptomyces albicans Aspergillus niger, and Trichoderma reesei, which can make the decomposition more complete. Mixing approximately 60 kg of compost in the bucket evenly. Next, divide the compost into six batches and start the control system. Once the high-temperature period is reached, collect the compost daily for measurement and statistical analysis using instruments. The aerobic fermentation process diagram for the composting equipment is illustrated in Figure 12.

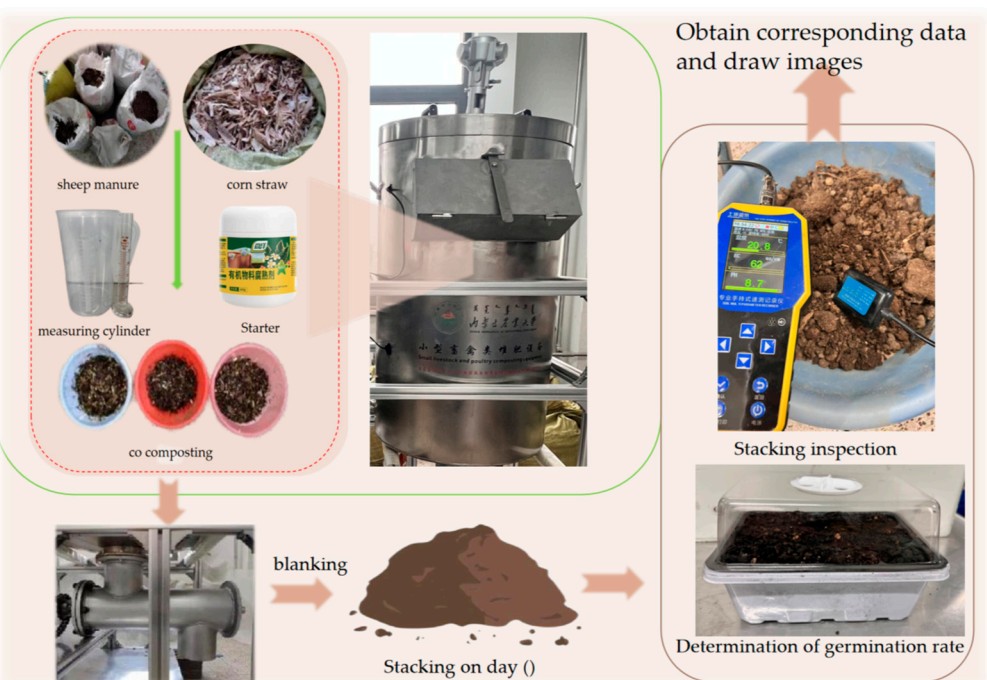

**Figure 12.** Schematic diagram of composting test process.

Once all the stacking materials are poured in, the changes in various parameters of the stacking materials and the air inside the tank can be observed through the touch screen. During the actual composting experiment, the temperature, humidity, oxygen concentration in the tank, air temperature in the tank, air humidity in the tank, and room temperature were recorded from 8 a.m. to 10 p.m. Data were recorded every hour until the end of the composting cycle.

When processing and analyzing the experimental data, it was noted that the changes in the stacking and air data in the tank did not occur in a transient process, meaning there were no significant parameter shifts in short intervals. As a result, it was decided

to use the data collected at 9 a.m., 3 p.m., and 9 p.m. daily as the objects for graph analysis. Considering that the temperature, humidity of the compost material, and oxygen concentration in the tank are the main factors affecting the compost material's ripening rate, they were selected as the primary analysis objects for the discussion of compost test results. The study of both unimproved equipment and natural composting has led to the conclusion that compost can mature after three rounds of aeration and heating [32]. The improved composting equipment also utilizes this three-round process to ensure reliable experimental comparisons.

After the compost starts operating, the appropriate conditions are reached, and the material is taken. Place a certain amount of stacked material in a pot for a certain number of days, use a soil sensor to detect it, and collect data. Finally, mix the stacked material with farmland soil, and put the corresponding number of mung bean seeds into the seedling pot. After the seeds germinate, the germination index of the seeds can be calculated based on the statistical data. Repeat the above experiment three times and take the average of the data obtained.

## 3. Result and Analysis

### 3.1. Temperature Changes in Composting

The temperature of a compost pile is a critical determinant of its quality. Sustaining a high temperature effectively eliminates pathogenic bacteria, thereby ensuring the safety and quality of the resulting compost product. As a demonstration, Figure 13 depicts the temperature fluctuations in the equipment both before and after improvements made during the composting process.

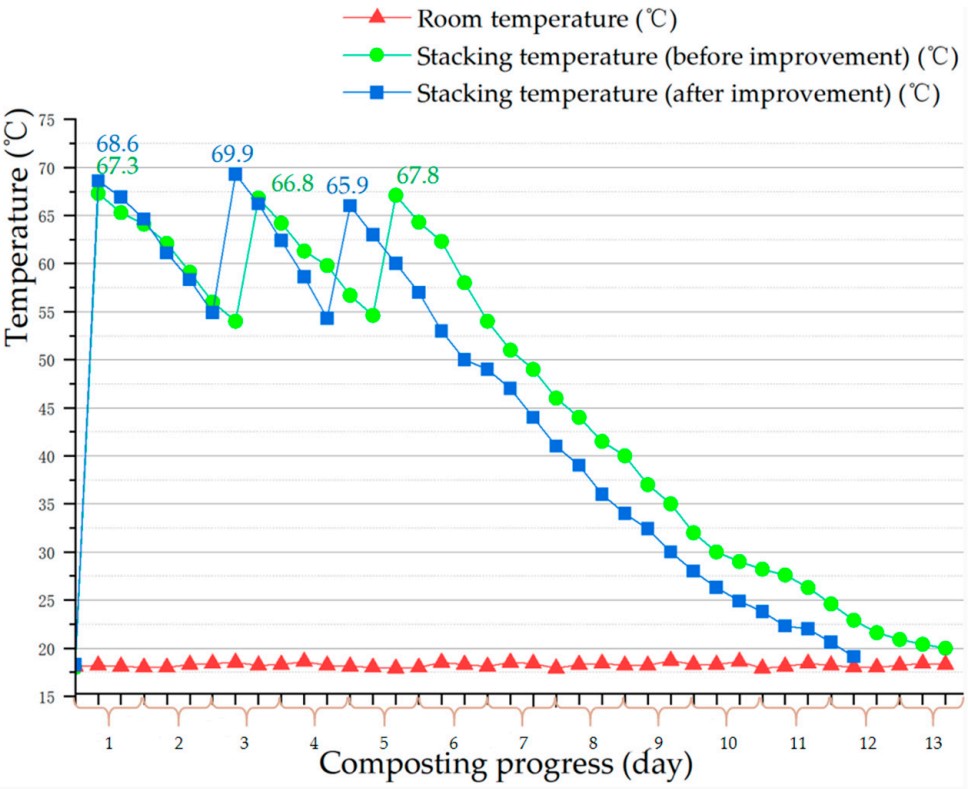

**Figure 13.** Comparison of temperature changes in composting.

According to the temperature change chart of the composting material, the total duration of this composting experiment is 11 and a half days, which is one day faster than composting and ripening without improved equipment. The high-temperature stage of composting under the ventilation and heating conditions of the improved and unimproved equipment lasts for about 5 and 6 and a half days, respectively. However, during the

subsequent cooling process, the temperature of the pile material drops below 40 °C after about 2 days. Compared to the natural fermentation composting experiment, this stage can still be considered the high-temperature stage of pile material fermentation [32]. Therefore, under the heating conditions of the improved composting equipment, the total duration of the high-temperature stage of composting is about 7 days, while under the heating conditions of the unimproved composting equipment, it takes about 8 days.

From the graph, it can be observed that the improved equipment starts the second ventilation only slightly earlier than the unimproved one, but there is a one-day difference in the time of the third ventilation. According to the control system's historical report, the improved composting equipment has a high degree of automation and initiates heating as soon as the temperature falls below 55 degrees Celsius. In contrast, unimproved equipment requires regular checks to ensure falling within the temperature range before making decisions, highlighting the efficiency of the improved composting equipment.

### 3.2. PH Value Changes in Composting

Once the artificial heating stage is completed, which begins on the 6th day of the composting process, the pH value of the compost material is sampled and tested. Figure 14 displays the changes in pH value that occur during the composting process. Based on the graph, it is evident that both before and after improvement, the pH value of the compost shifts from weakly alkaline to weakly acidic during the middle and later stages of composting. This observation aligns with the findings of natural fermentation composting experiments. At the completion of composting, the modified pH value is 6.6, which meets the industry standard range of 5.5 to 8.5.

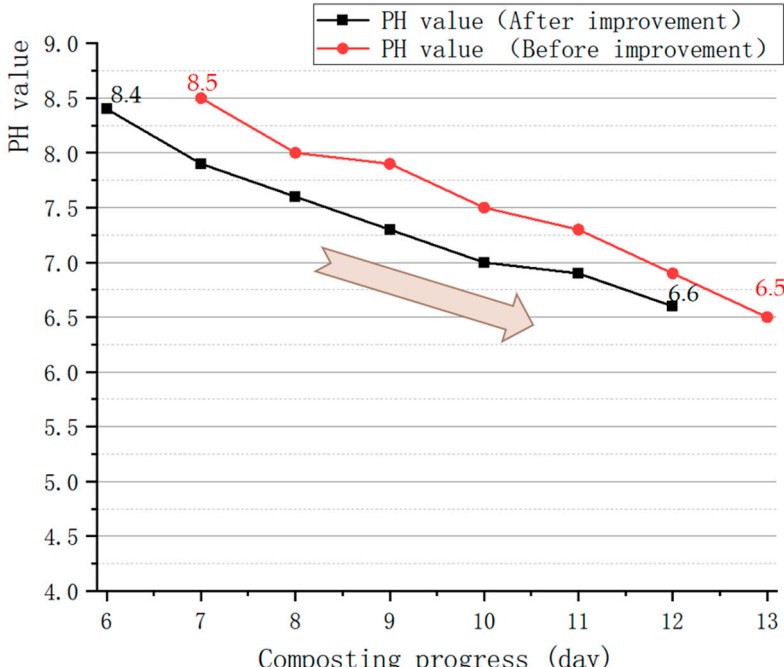

**Figure 14.** Test result of pH value.

### 3.3. Oxygen Concentration Changes in Composting

Oxygen concentration is a crucial indicator in the composting process. The optimal oxygen concentration during this process is 15~20%. When the oxygen concentration is too low, some areas may become anaerobic, limiting the activity of aerobic microorganisms and potentially producing harmful gases. According to the oxygen concentration change curve in Figure 15, the oxygen concentration in the tank gradually decreases due to the composting material's fermentation but increases again after each ventilation.

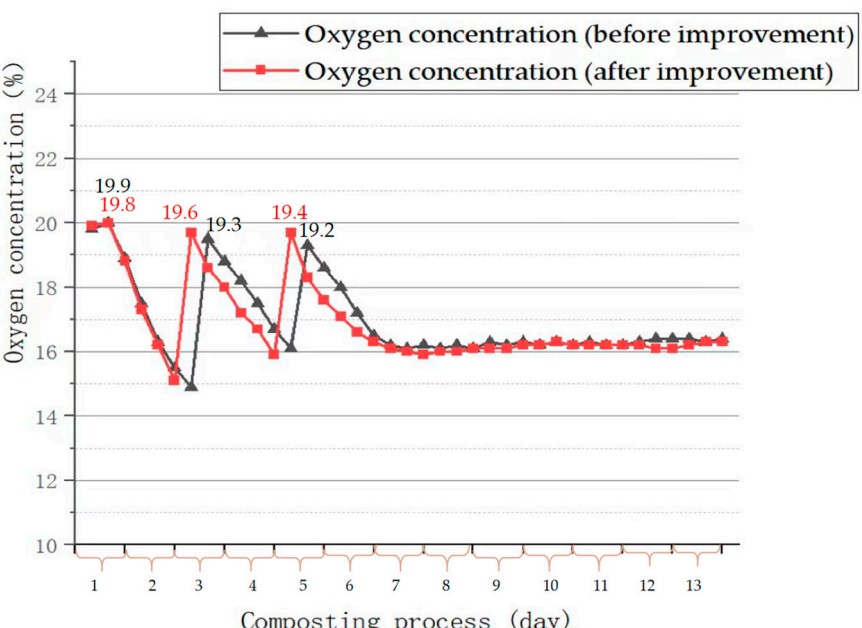

**Figure 15.** The change of oxygen concentration in the tank.

From Figure 15, it can be seen that the improved oxygen consumption from 20% to about 15% took about 1 and a half days each time, while it took about 2 days before the improvement. From the oxygen consumption rate, it can be seen that the improved equipment is more efficient. Since the composting material is mostly matured during the cooling stage, its oxygen consumption is minimal. The fermentation tank's semi-enclosed form contributes to a slight increase in the tank's oxygen concentration after composting. As the composting process unfolds, aerobic fermentation takes place within the pile, necessitating the consumption of a substantial amount of oxygen. Oxygen is consumed more rapidly during the initial stages of composting. As fermentation of the compost advances, however, the rate of oxygen consumption progressively decreases. Ultimately, when the compost matures, the pile essentially stops consuming oxygen.

### 3.4. Humidity Changes in Composting

Figure 16 illustrates the variation in humidity throughout the composting process. The decline in compost humidity, both before and after improvements, follows a similar pattern; it continuously descends, more noticeably during the high-temperature stage, then gradually decreases at a slower pace during the cooling stage, ultimately stabilizing at around 9%. Noteworthy, however, is the accelerated decline in humidity during the high-temperature phase after improvements were made. This can be attributed to the enhanced mixing effectiveness of the improved equipment, which enables the compost to absorb more heat and consequently speeds up water loss.

### 3.5. Seed Germination Index Changes in Composting

The Germination Index (GI) is an indicator used to evaluate the degree of harmless and stable composting by assessing the compost's inhibitory effect on plant germination. This index can detect the level of plant toxicity in compost samples and predict changes in plant toxicity within the compost. As composting progresses, microbial degradation of materials produces macromolecular substances that inhibit plant growth. Over time, thermophilic microorganisms break down these macromolecular substances into smaller molecules, leading to a gradual decrease in the materials' inhibition of seed germination.

According to Figure 17a, the germination index (GI) displays a significant increase during composting. At the end of the high-temperature period, both the unmodified and improved compost had GI values near 40%. However, following composting, both types of

compost exhibited GI values near 90%. The improved composting process resulted in a GI increase of 89%, which is significantly higher than the 83% increase observed in unmodified compost. This suggests that the degree of composting maturity was greatly enhanced by the improved process. As depicted in Figure 17b, the error bar analysis of the seed germination rate from the compost produced by the improved equipment reveals a small error value. This suggests a high degree of accuracy in the data recorded by the equipment.

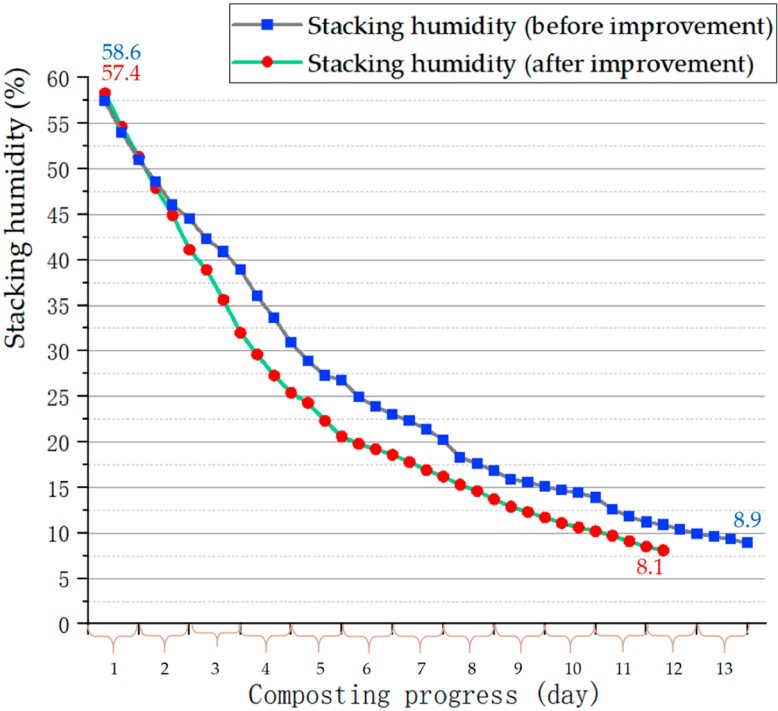

**Figure 16.** The change of composting materials humidity.

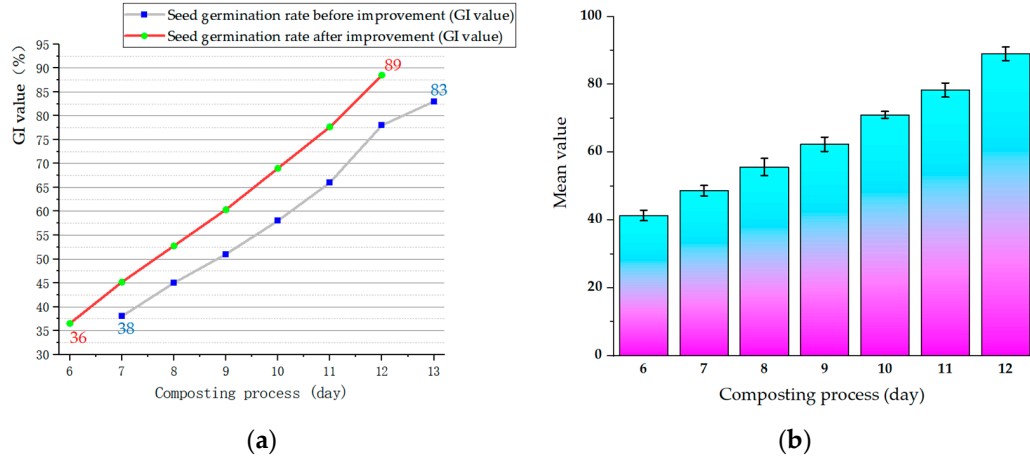

(**a**)                    (**b**)

**Figure 17.** Test result of seed germination index. (**a**) Changes in germination rate before and after improvement; (**b**) Improved Error bar diagram of germination rate.

### 3.6. Comparison of Test Results and Evaluation of Equipment Practicality

The composting effects of the equipment before and after improvement are shown in Table 2, where the effectiveness indicators obtained from the experiment are evaluated. The Chinese standard document "Technical Specification for Harmless Treatment of Livestock and Poultry Manure" (GB/T 36195-2018) stipulates that solid livestock and poultry manure treatment must be maintained at a high temperature for no less than 7 days. The improved equipment's high-temperature duration meets this national standard at 7 days.

**Table 2.** Comparison of experimental results.

| Serial Number | Test Parameters | Unimproved Equipment | Improved Equipment |
|---|---|---|---|
| 1 | Composting cycle | 13 days | 11 and a half days |
| 2 | High-temperature maintenance time | 8 days | 7 days |
| 3 | Maximum temperature of stacking material | 67.8 °C | 69.9 °C |
| 4 | Minimum humidity of stacking material | 8.9% | 8.1% |
| 5 | Maximum Seed Germination Index (GI) | 83% | 89% |
| 6 | Minimum pH value | 6.5 | 6.6 |
| 7 | Minimum oxygen concentration | 14.8% | 15% |

According to the industry standard document "Organic Fertilizers" (NY 525-2012), the moisture content of composted organic fertilizers should be less than 30%, and the pH should range between 5.5 and 8.5. The experimental results demonstrate that the organic fertilizers produced under both conditions fulfill these requirements for moisture content and acidity.

The composting experiment also uses the Seed Germination Index (GI) as an auxiliary indicator for evaluating fertilizer maturity. When GI > 80%, it indicates that the compost has reached maturity and is essentially free from plant toxicity. Although the GI value of the fertilizer produced before improvement is about 83% and meets the requirements, under the improved conditions, the GI value increases to around 89%. This increased value aligns more closely with the low-plant-toxicity requirement for organic fertilizers.

Analyzing and comparing the results of two composting experiments (Table 2) reveals that the optimized composting equipment discussed in this article effectively increases the compost material's fermentation temperature and shortens the composting cycle. This improvement enhances the composting efficiency of the material and addresses the issues of poor mixing effect and low automation level found in traditional composting equipment. Additionally, the organic fertilizer produced by this equipment exhibits a higher degree of maturity than that derived from natural fermentation compost. The thorough decomposition of the organic matter in the compost also suggests that this type of organic fertilizer has less harmful effects on crops.

**4. Discussion**

This study was designed to investigate the efficiency of improved composting equipment in enhancing the composting process of livestock manure. The optimized equipment addresses issues of poor mixing effect, low automation level, and extended composting cycles in traditional composting methods. The analysis of the experiment results, comparisons with industry standards, and evaluation of the maturity of produced organic fertilizer offer valuable insights into the optimized composting equipment's efficiency and practicality. Our team conducted preliminary research on the design, processing, and construction of various parts of the device, laying a solid foundation for subsequent research.

The improved composting equipment successfully maintained the compost material's high-temperature duration at 7 days, which is in compliance with the Chinese standard document "Technical Specification for Harmless Treatment of Livestock and Poultry Manure" (GB/T 36195-2018). The high-temperature maintenance period of compost is slightly shorter than before improvement because the full-range stirring and stacking of materials maintain the temperature inside the fermentation tank at a relatively average state, with most stacking materials maintaining the temperature. However, unimproved equipment has some areas that cannot be stirred, leading to poor temperature absorption of the stack material. Consequently, the improved blades consume heat faster than the unmodified ones, resulting in earlier ventilation and heating. This observation further demonstrates that the improved mixing blades have better effects.

Additionally, the organic fertilizers produced under both the improved and unimproved conditions met the moisture content and acidity requirements outlined in the industry standard document "Organic Fertilizers" (NY 525-2012). Such compliance with

industry standards highlights the equipment's ability to speed up the composting process while still producing high-quality compost.

Further, the Seed Germination Index (GI) was used to evaluate organic fertilizer maturity, indicating a higher degree of maturity and less potential harm to crops with fertilizers produced from the improved composting equipment. The GI value increased from 83% to 89% with the improved conditions, demonstrating the optimized equipment's capability to improve the overall quality of the organic fertilizer.

The upgraded composting equipment's improved mixing structure, ventilation heating system, and automation level ultimately contributed to the enhanced efficiency of livestock manure composting and increased the utilization value of the resulting organic fertilizer. Our findings verify the feasibility and practicality of optimizing composting equipment through practical composting experiments. In addition, the composting process's shortened duration, combined with the higher degree of finished organic fertilizer's maturity, demonstrates the potential for optimized composting equipment to boost agricultural efficiency and sustainability.

Despite the successful outcomes, this study has several limitations that warrant consideration. First, the composting experiments were conducted using a specific type of livestock manure—sheep manure—with corn straw. Whether this equipment can achieve good results in composting other livestock manure and other types of organic matter has not been tested yet. Second, the impact of external factors, such as climate conditions, on the improved equipment's performance should be further investigated.

In conclusion, optimizing composting equipment by enhancing its mixing structure, ventilation heating system, and automation level has the potential to significantly improve the composting process of livestock manure while generating high-quality organic fertilizer. The practical composting experiments conducted in this study validated the feasibility and practicality of such optimizations.

In the past, soil conservation efforts have involved the use of materials such as earthworm charcoal and biochar, which were mixed to enhance soil structure and supply assimilable nutrients for plant growth [43,47–49]. The research presented in this paper further bolsters the development of sustainable agricultural practices. It addresses the escalating global demand for organic fertilizer, mitigates the environmental issues associated with livestock manure management, and introduces a novel approach to soil conservation. Furthermore, efficient management of livestock and poultry manure also plays a crucial role in the scalability of breeding farms. Free from the constraints of manure accumulation, these farms could expand significantly. Such a development would substantially propel the growth of China's aquaculture industry.

## 5. Conclusions

This study's findings demonstrate the potential of optimized composting equipment in significantly enhancing the composting efficiency of livestock manure and producing high-quality organic fertilizer. The key conclusions that can be drawn from our analysis include:

1.  Enhancements in the mixing structure, ventilation and heating system, and automation level greatly contribute to the overall efficiency of the composting process. These improvements could potentially have a significant impact on agricultural sustainability and address challenges related to livestock manure management.
2.  The seed germination index (GI) value of the organic fertilizer produced using the improved equipment has increased from 81% to 89%, suggesting a higher maturity level and less potential harm to crops. The pH value remains within the industry standard of 5.5–8.5, recorded specifically at 6.6. The minimum stacking humidity has been reduced slightly from 8.9% to 8.1%, still well within the acceptable industry standard of less than 30%.
3.  The modified composting equipment successfully maintained the high-temperature duration of the composting material for a period of 7 days, complying with industry regulations.

**Author Contributions:** Conceptualization, methodology, writing—original draft preparation, and investigation, K.R. and L.S.; validation, formal analysis, and writing—review and editing, X.H. and X.C.; and writing—review and editing, supervision, Y.Z. All authors have read and agreed to the published version of the manuscript.

**Funding:** This research was funded by The Innovation Team of Higher Education Institutions in Inner Mongolia Autonomous Region, grant number NMGIRT2312 and The Natural Science Foundation of Inner Mongolia Autonomous Region, grant number 2022QN03019.

**Institutional Review Board Statement:** Not applicable.

**Informed Consent Statement:** Not applicable.

**Data Availability Statement:** Not applicable.

**Conflicts of Interest:** The authors declare no conflict of interest.

## Abbreviations

| | |
|---|---|
| MCGS | Monitor and Control Generated System |
| PLC | Programmable logic Controller |
| $^{\circ}C$ | Anders Celsius |
| kJ | Kilojoules |
| h | hour |
| K | Kelvins |
| $m^3$ | cubic meter |
| mm | millimeter. |
| W | Watt |
| GI | Germination index |
| PH | Potential of hydrogen |
| $q_{eht}$ | Heat released by the electric heating tube |
| $q_{ia}$ | The heat absorbed by the air pump during heating |
| $q_{hc}$ | Heat conduction on the wall of the heating box |
| $q_{ste}$ | Heat absorbed by water evaporation in the air |
| $q_{else}$ | Other forms of heat loss |
| $c_{air}$ | Specific heat capacity of air |
| $\rho_{air}$ | Air density |
| $\phi_{air}$ | Ventilation capacity |
| $\Delta T_{air}$ | Air temperature change value |
| $\lambda_{air}$ | Air thermal conductivity |
| $A$ | Contact area |
| $\Delta T$ | Temperature difference between objects |
| $d$ | Heat transfer distance |
| $\Delta x$ | Net mass of water vapor in the air |
| $q_l$ | Enthalpy value of water vapor |

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
