# Peer review of "Optimization and Experiment of Livestock and Poultry Manure Composting Equipment with Vented Heating"

_sustainability, doi:10.3390/su151411353_

Round 1

Reviewer 1 Report

Authors focused on optimization of small-scale composting systems. They redesigned the mixing blades and refined the ventilation heating system, establishing an efficient mixing mechanism and an energy-saving ventilation heating system. A control system, incorporating MCGS and PLC, was developed for real-time monitoring and adjustment capabilities, substantially enhancing automation levels. The subject matter is extremely important nowadays, as it helps to minimize the generation of fecal waste.

Comments and Suggestions for Authors

·         Line 14: MCGS and PLC - there should be an expansion of the name in parentheses.

·         In the caption under Figure 1 when listing components after the numbers there should be a space.

·         Figures 11 and 15 - blurred lettering.

·         Line 390: missing space before parenthesis [38-41].

·         Line 460: missing space before parenthesis [42-45].

·         Line 483: missing space before parenthesis [31].

·         The Conclusion section is too short - can it be expanded?

·         Uniform spaces before specifying units - once spaces are inserted before units once not.

·         No comparison with other existing devices - without this, it is impossible to determine whether the device works better or not.

·         Can the equipment be enlarged because it is small-scale? Will the device retain its advantages after being upsized?

Author Response

请参阅附件。

Reviewer 2 Report

In this article, the authors redesigned and optimized a composting system and integrated with MCGS and PLC control modulus. The optimized system was demonstrated to provide efficient composting process and result in high quality. Please see my comments below.

1. Line 134. Is "heat" a typo? Seems "air" should be used here.

2. The 4th paragraph in page 5 is hard to understand. It first says that the heat loss of heating box due to temperature difference. Then it cites an article to state that fermentation tank contributes significant heat loss. Is this heat loss by fermentation tank calculated in Eq. (1)? 

3. I have one concern regarding Eq. (1) if I am not messed myself up with thermal physics. Does the water go through phase change during the heating process? If not, why calculate q_ste? Although based on the calculation, q_ste term has almost no contribution to q_eht.

4. Can the authors quantitatively analyze the energy-saving quality of implementing insulation measures on the heating system? For example, energy saved during pre-heat process. Evaluation using time (2 seconds faster) seems not straightforward.

5. Line 252. Typo. It should be "Figure 5" instead of "Figure 6".

6. Line 285. Similarly, it should be "Figure 7" instead of "Figure 8".

7. For DEM simulation, what was the particle size used and how were parameters, such as surface energy, coefficients of friction and restitution determined? Any calibration work performed?

8. Please improve the written English. For example, in line 515, it's inappropriate to say "absorbing temperature", but "maintaining temperature" sounds more proper. 

9. Why is the optimal oxygen concentration is 18% specifically during the composting process? Does the improvement help with achieving optimal oxygen level in the reactor? Similar concern regarding pH and humidity level.

Written English needs to be improved. Please carefully do the proof reading as there are a lot of typos in the text.

Reviewer 3 Report

The paper was revised according to the journal rules. The topic treated was focused on the optimization of poultry manure composting.

Few revisions are required and they are reported below:

- a nomenclature list with acronyms and parameters must be added, please add also SI unit of measure

- in the introduction section, the successful decomposition may be assessed with special techniques, such as direct injection measuring techniques to check the status of degradability, you could considers few articles as reference,  such as (10.1016/j.ijms.2018.05.002, 10.1016/j.biortech.2011.01.016, 10.17660/ActaHortic.2008.791.111)

- please check that all details required by the instruments reported in section 2 are properly added

- figure 4 could be improved

- please use SI for the unit of measure

- figure 9 could be improved

- the particle structure and porosity of the treated biomass should be deeply treated

- figure 13 could be improved

- maybe some figures can be moved to a supplementary material section

- figure 16,19 and 20 could be improved

- being an experimental study, the uncertainty analysis section can be added

- please include the uncertainty bars, the variation bars are required

- the discussion section can be completed adding few already published results to support the work accomplished

minor revisions

Author Response

请参阅附件

Reviewer 4 Report

The author has improved the efficiency and automation of the small-scale composting system by optimizing it. However, there are currently serious issues with the manuscript, mainly including that it seems not a research paper but rather a report. Therefore, it is recommended that the author make major revisions to the entire text. 

1. The manuscript is more like a report, without clear research objectives, content, and results; 1. It is recommended to use section 2 as an experiment and method, mainly describing the introduction of experimental equipment, differences before and after optimization, and methods for measuring experimental results, etc., to simplify the existing content. 2. It is recommended to add comparative data analysis of device results before and after renovation in sections 3; 3. The result discussion section of the experimental results requires a further in-depth analysis of the differences in the results; 4. The conclusion section lacks key data, and it is recommended to revise it. 

2. All abbreviations in the manuscript need to be marked with their full names when they first appear. 

3. The format of the references is not uniform. At the same time, please add the latest research-related references, and use journal references instead of master's or doctoral theses.

Suggest the author revise the manuscript.

Round 2

Reviewer 4 Report

 Accept.

Author Response

Dear reviewer:

        Thank you for your decision and constructive comments or my manuscript. Thank you for your affirmation of our resubmission of the article. Once again, thank you very much for your comments and suggestions.

        Wishing you a happy life! Work smoothly!